# Terminal Crossbreeding of Murciano-Granadina Goats with Boer Bucks: Characteristics of the Carcass and the Meat

**DOI:** 10.3390/ani12192548

**Published:** 2022-09-23

**Authors:** Ión Pérez-Baena, Cristòfol Peris, Nemesio Fernández, Jorge Franch-Dasí, Cristhian Sagbay, Julio Cesar Terán-Piña, Martín Rodríguez

**Affiliations:** 1Institut de Ciencia y Tecnología Animal, Universitat Politècnica de València, Camí de Vera s/n, 46022 Valencia, Spain; 2Globalgen Institut, Universidad Politécnica Salesiana, Calle Vieja 12-30 y Elia Liut, Cuenca 010105, Ecuador; 3Departamento Académico Agrosilvopastoril, Universidad Nacional de San Martín, Jr. Maynas 177, Terapoto 22200, Peru

**Keywords:** suckling kids, carcass quality, meat quality, Murciano-Granadina, Boer

## Abstract

**Simple Summary:**

In Murciano-Granadina-breed dairy goat farms, meat production from purebred kids is not very profitable due to its high costs, so it is proposed to cross unused dairy females to produce a replacement for the herd with a breed specialised for meat production, i.e., the Boer breed. Our previous results verified that this crossbreeding strategy improved the productive characteristics of the kids (growth and costs), so the aim of this work was to ascertain its consequences on the carcass and meat characteristics of suckling kids (9 kg weight at slaughter) of both sexes. Ninety-four kids were used. It was found that crossbred kids reached slaughter weight at a younger age, and although their carcass yield did not improve, they did show higher muscle/bone and meat/bone ratios and better meat texture (less firmness). In turn, males slightly improved the proportion of expensive cuts, but their carcasses contained a lower proportion of intermuscular fat and lower muscle/bone and meat/bone ratios than females. There were no sensory differences in any case. It is concluded that this crossbreeding also improves the carcass and meat characteristics, constituting a valid alternative to improve the economic results of Murciano-Granadina breed herds.

**Abstract:**

After verifying productive improvements from the crossbreeding (MB) of Boer males with females of the Murciano-Granadina (MG) breed, the aim of this work was to determine its consequences on the carcass and meat characteristics of suckling kids (9 kg of weight at slaughter) of both sexes. A total of 94 kids (25 purebred MG males, 19 purebred MG females, 25 crossed MB males and 25 crossed MB females) were used. It was found that MB kids reached slaughter weight at a younger age (44 ± 1 vs. 63 ± 1 days, *p* < 0.001). For carcass yield, the interaction genotype x sex was observed, where MB females presented a higher value (51.71 ± 0.34%) than the rest of the animals (MB males 50.53 ± 0.34, *p* < 0.05; MG males 50.60 ± 0.34, *p* < 0.05; MG females 49.62 ± 0.39, *p* < 0.001). MB kids had greater leg compactness, (36.28 ± 0.27 vs. 34.71 ± 0.29 g/cm, *p* < 0.001), slightly improved expensive carcass cuts (75.93 ± 0.31 vs. 74.77 ± 0.34%, *p* = 0.014) and higher muscle/bone (2.93 ± 0.03 vs. 2.76 ± 0.03, *p* < 0.001) and meat/bone ratios (3.56 ± 0.04 vs. 3.36 ± 0.04, *p* < 0.001) than MG kids. They also showed a somewhat more intense red colour (8.57 ± 0.25 vs. 7.74 ± 0.27, *p* = 0.027), higher colour saturation (9.06 ± 0.24 vs. 8.20 ± 0.26, *p* = 0.016) and less meat firmness (1.95 ± 0.08 vs. 2.23 ± 0.081 kg/cm^2^, *p* = 0.021). On the other hand, males had a slightly improved proportion of expensive cuts (75.98 ± 0.31 vs. 74.73 ± 0.34, *p* = 0.008), but their carcass contained a higher proportion of bone (22.16 ± 0.21 vs. 21.17 ± 0.17%; *p* < 0.001), a lower proportion of intermuscular fat (9.54 ± 0.24 vs. 10.93 ± 0.24, *p* < 0.001), a higher muscle/fat ratio (5.01 ± 0.10 vs. 4.41 ± 0.10, *p* < 0.001) and lower muscle/bone (2.80 ± 0.03 vs. 2.89 ± 0.03%, *p* = 0.016) and meat/bone ratios (3.36 ± 0.04 vs. 3.56 ± 0.04%, *p* < 0.001) than females. There were no sensory differences in any case. It is concluded that this crossbreeding strategy also improves the carcass and meat characteristics, making it a valid alternative to improve the productive results of MG dairy goat herds.

## 1. Introduction

Goats are distributed throughout the world due to their ability to adapt to different environments and to provide quality food, so they are considered the main supplier of meat and milk for rural populations worldwide [1]. At present [2], this type of animal is basically used for meat production, as 80.32% of the world domestic goat population is used for this production purpose, although in the European Union (EU), 73.4% of the population is used for milk production. So, the EU, with 12.2 million head (1.1% of the world domestic goat population), contributes 12.3% of world goat milk production and 36.7% of goat cheese production [2]. More specifically, the Mediterranean Basin countries produce 73.14% of the EU’s goat milk, as it is an area with a long tradition of small-ruminant milking [3]. In Spain, the breed of goats most frequently used is the Murciano-Granadina (MG), which has a domestic goat population of about 500,000 animals [4], with an average milk production of 547 kg in 210 days of lactation, containing 5.47% fat and 3.52% protein [5]. It has an adult size of 45–55 kg in females and 60–70 kg in males [6], with good average prolificacy (2.1), low birth weight (2.25 kg) and a low growth rate during lactation (114 g/day) [7]. The breed has a high dairy aptitude, but its qualities for meat production are more limited, so the sale of kid meat represents around 10% of the total income of the farm and may not even compensate for the costs involved [8].

An alternative used to improve meat production in goats consists of crossing the local breed with Boer males, which is a breed that enjoys worldwide recognition for its excellent body conformation, rapid growth rate and good carcass quality [9]. In some large breeds, such as the Kiko, crossing with the Boer breed does not improve the growth of kids [10], but in the MG breed, Pérez-Baena et al. [7] showed that crossbreeding improved birth weight, the survival of offspring and the growth and conversion rates of kids during lactation, so production costs are reduced. Consequently, these authors considered the crossbreeding of MG females with Boer males to be of interest for the production of suckling kids. One characteristic of the goat meat market in southern European countries, including Spain, is the consumer preference for low-weight (7–9 kg) and young (1–2 months) suckling kid carcasses [11]. However, most of the available references are related to crossed kids that are not usually slaughtered at such an early age and weight [12,13,14,15,16,17,18,19,20,21,22]. Given that our group has verified that crossbreeding MG females with Boer males improves the productive characteristics of the kids, the aim of this work was to determine the possible consequences that it may have on the carcass features and the meat of kids with low slaughter weight.

## 2. Materials and Methods

### 2.1. Goats and General Procedures

In a previous work (Pérez-Baena et al. [7]), at the experimental farm of the Spanish Universitat Politècnica de València, 70 multiparous (2.8 ± 0.2 years) healthy Murciano-Granadina (MG) goats, weighing 47 ± 1.5 kg on average, were divided into two groups of 35 animals of similar age and body condition (2.5–3). One of the groups was randomly assigned to mate with MG males (n = 4), producing purebred kids (MG), and the other group was assigned to mate with Boer males (n = 4), producing crossed Murciano-Granadina × Boer (MB) kids. The goats remained housed and were managed together under the same environmental and feeding conditions. Ninety-four kids of 4 types (KT) were used: MG males (25), MG females (19), crossed MB males (25) and crossed MB females (25). The handling of the animals and the productive results can be consulted in [7]. All of these kids were used in this experiment to study the performance of the carcass, its morphological characteristics, the cutting of the carcass, the Longissimus thoracis et lumborum (LTL) muscle, the pH and the meat colour, but for the study of the tissue composition of the carcass and other meat characteristics, including a sensory analysis, only 15 randomly selected animals of each type were used.

### 2.2. Slaughter and Carcass Management

The animals were slaughtered in a certified commercial slaughterhouse. The carcass was defined according to [23] and did not include the head or viscera (lungs, heart, liver and spleen) but included the tail, kidneys, and renal and pelvic fat. The testicles and scrotal fat were also removed. Slaughter age, slaughter live weight (SLW), hot carcass weight (HCW), cold carcass weight (CCW) and the weights of components that are not part of the carcass (NCWC), namely, blood, skin, feet, head, complete gastrointestinal tract and viscera, were recorded. The carcasses were hung by the Achilles tendon and refrigerated for 24 h at 4 °C. The yields of the hot (HCY) and cold carcasses (CCY) were calculated as: HCY = ((HCW / SLW) × 100) and CCY = ((CCW/SLW) × 100).

### 2.3. Measurements of the Carcass

According to the methodology described by [24], after 24 h of carcass cooling, the croup perimeter and croup width were measured. Measurements of the leg length, the internal length of the carcass and the depth of the thorax were taken on the suspended left half canal from the Achilles tendon, which allowed calculating the compactness index of the carcass as the ratio between the weight of the cold carcass and the internal length of the carcass.

On the transverse section of the LTL muscle (13th thoracic vertebra), the major diameter (A), the minor diameter (B) and the thickness of the subcutaneous fat were measured. The rib-eye area was calculated using the geometric procedure [Surface = (A/2) × (B/2) × π] according to [25], and the muscle shape index was measured as the ratio between the largest diameter and the smallest diameter [(B/A) × 100].

Carcass cutting was performed on the left half of the carcass, separating the five primary cuts (leg, shoulder, rib, flank and neck) and the three minor cuts (kidney, kidney fat and tail) described by [23]. These cuts were weighed, and the results were expressed as a percentage of the weight of the half-carcass. The leg compactness index was calculated as the ratio: leg weight/leg length.

The left shoulder was used to study the tissue composition of the carcass, following the methodology of [26]. After cutting, the shoulder was vacuum-packed and kept frozen at −24 °C until dissection. Thawing was carried out at a temperature of 4 °C for 24 h. With the help of forceps and a scalpel, the tissues were dissected: subcutaneous fat, intermuscular fat, muscle, bone and the remains or debris, which includes blood vessels, ligaments and tendons. The results were expressed as a percentage of the shoulder weight.

### 2.4. Meat Characteristics

The pH was evaluated (mean of three measurements) on the right half of the carcass in the LTL muscle by means of a penetration electrode (Hanna HI 99163; Hanna Industries. Eibar, Spain) inserted between the 12th and 13th thoracic vertebrae after slaughter (pH0), at 45 min (pH45) and after 24 h of refrigeration (pH24).

The colour was determined after 24 h of chilling the carcass. It was measured in the LTL muscle at the level of the 13th vertebra (after taking a cross-section of the muscle and 30 min of oxygenation) and in the Rectus abdominis muscle with a portable CR300 Minolta colorimeter (Valencia, Spain), CIELAB space, Illuminant D65 and an observation angle of 10°. The luminosity (L*), red index (a*) and yellow index (b*) were recorded. From these, saturation (C*) was calculated as C* = (a*^2^ + b*^2^)^0.5^, and tone (H^o^) was calculated as H^o^ = arctg (b*/a*), expressed in degrees. The results were expressed as the average of three measurements on the same surface.

To study the water retention capacity (WRC) and cooking losses (CL), the methodology described by [27] was followed on samples of the LTL muscle, between the 10th and 13th vertebrae. For the WRC, the percentage of juice released by compression was evaluated 24 h after slaughter. Cooking losses were evaluated in samples thawed and cooked in a water bath at 75 °C for 30 min. A texture analysis was carried out on the cooked meat sample. Parallelepipeds of 2 × 1 × 1 cm (2 pieces per sample) were cut, following the direction of the muscle fibres. The texture was assessed using a Warner–Bratzler cell coupled to a model TA-XT2 texturometer (Hamilton, IL, USA) with a speed of 5 mm/s, which made complete cuts in the parallelepipeds. The maximum force (kg/cm^2^) was obtained for a complete cut of the sample [28], with the firmness of the shear (kg/cm^2^) being defined as the slope from the origin to the point of maximum force, and the area under the curve (total work) being obtained as the work necessary for the complete cut of the sample (kg/cm^2^) [29].

The study of the chemical composition of the meat was carried out (in duplicate) on a sample of the LTL muscle between the 6th and the 10th thoracic vertebrae, which was vacuum-packed and frozen at −24 °C until its analysis. Once thawed, the analytical procedures of the official methods were followed: moisture [30], protein [31], fat [32] and ashes [33].

### 2.5. Sensory Evaluation of Meat

The sensory evaluation of the meat was carried out by means of 3 types of tests: a panel of expert judges, a controlled consumer hedonic test and a family hedonic test.

The panel of expert judges was carried out according to [34] using standardised cabinets with red light and a panel of 7 expert judges from the Zaragoza Veterinary Faculty. The part of the LTL muscle from the lower back of the left half of the carcass was used. The samples were thawed for 24 h under refrigeration and then cooked on a double-contact grill plate at 200 °C until the inside of the meat reached a temperature of 70 °C. Once cooked, they were cut into 2 × 2 cm pieces, identified by 3 digits on aluminium foil and tasted within a period of less than 10 min after cooking. Each judge tasted 60 samples (15 from each KT) in 5 sessions, with 3 samples from each KT per session, and evaluated 12 sensory attributes per sample (goat odour, fat door, tenderness, juiciness, stringiness, amount of residue, kid flavour, liver flavour, metallic flavour, acid flavour, persistence of flavour and general acceptance). The panellists used a 100-point scale to assess the different sensory variables.

The controlled consumer hedonic test was carried out following [35] in the Instituto de Ciencia y Tecnología Animal tasting room, equipped with booths with white light. The part of the LTL muscle of the lower back of the right half of the carcass was used, and 56 tasters participated (58.9% men and 41.1% women), aged 23 to 29 years (65%) and 30 to 60 years (35%). The samples, once thawed, were cut into cubic pieces 1.5 ± 0.2 cm thick, wrapped in aluminium foil and identified by 3 digits. Subsequently, they were cooked as described for the panel of judges. Each taster tasted 4 samples, one from each KT. According to [34], the tasters evaluated the degree of acceptance using a structured hedonic scale of 15 cm, which ranged from 1 (I dislike it extremely) to 15 (I like it extremely).

For the hedonic family test, the shoulders and ribs of the right half of the carcass and the legs of the carcass were used. It was carried out in 30 families (10 for each type of cut), with 135 tasters (65 men and 70 women), aged less than or equal to 29 years (37.8%), from 30 to 60 years (48.9%) and over 60 years (13.3%). Each family baked 4 samples of a carcass cut (one from each KT). In turn, each piece had its respective cooking bag for the oven with a numbered flange. The assessment was carried out following the standard [35].

### 2.6. Statistical Analysis of Results

For the statistical analysis of the data, the GLM procedure of [36] was used with a model that included the fixed effects of genotype, sex and their interaction. With the effects that were significant (with more than two levels), a pairwise comparison of means was performed by applying Student’s *t*-test.

However, the models used for the variables of the sensory evaluation of meat were as follows:

(1) Panel of expert judges. The model included the fixed effects of the genotype, sex, genotype x sex interaction and session (n = 5) and the random effects of the judge (n = 7) and kid sample (n = 60).

(2) Controlled consumer hedonic test. The model included the fixed effects of the genotype, sex and genotype x sex interaction and the random effect of the judge (n = 56).

(3) Hedonic family test. The model included the fixed effects of the genotype, sex, genotype x sex interaction and the type of cut (shoulder, rib and leg) and the random effects of the family (n = 30) and judge (n = 86).

For the last three models, the MIXED procedure of [36] was used.

## 3. Results

### 3.1. Obtaining the Carcass

Table 1 shows the age and weight of the kids at slaughter and NCWC. It can be observed that the genotype x sex interaction was not significant in any case except for the full gastrointestinal tract (*p* = 0.003). MB kids reached the slaughter weight at a younger age than MG kids, and males reached it more quickly than females (*p* < 0.001). The genotype influenced some variables from NCWC, such as the weights of the skin and of the feet (*p* < 0.001), which were higher for crossbred kids than for pure MG. The sex of the kids affected the weights of the feet (*p* < 0.001), head (*p* = 0.006) and visceral organs (liver, lungs and heart; *p* = 0.019), which were higher in males than in females. In the case of the observed interaction, the weight of the gastrointestinal tract in males was similar (*p* = 0.418) in MB (1224 ± 39 g) and MG kids (1269 ± 39 g), but in females, this weight was greater (*p* < 0.001) in MG (1508 ± 45 g) than in MB (1269 ± 39 g).

Table 2 shows total NCWC, the carcass weights for both HCW and CCW, and carcass yields for both HCY and cold CCY from purebred MG and crossed MB kids. The interaction genotype x sex was significant in all cases. In males, NCWC weights (*p* = 0.463), carcass weights (HCW and CCW, *p* = 0.529 and *p* = 0.855, respectively) and carcass yields (HCY and CCY, *p* = 0.502 and *p* = 0.882, respectively) did not differ between genotypes, but in females, it was observed that MB kids had lower NCWC (*p* = 0.020), higher carcass weights (HCW and CCW, *p* < 0.001 and *p* < 0.001, respectively) and higher carcass yields (HCY and CCY, *p* < 0.001 and *p* = 0.001, respectively) than MG kids.

### 3.2. Morphological Measurements of the Carcass

The results of the objective measurements taken on the carcass are shown in Table 3. The interaction genotype x sex was not significant in any of the cases except for the carcass compactness index (*p* = 0.006). It was verified that the genotype affected the croup width (*p* = 0.002), leg length (*p* < 0.001), chest depth (*p* < 0.001) and leg compactness index (*p* < 0.001), observing a genotype x sex interaction in the carcass compactness index. MB kids showed a smaller croup width, leg length and chest depth than MG kids. The carcass compactness index for male kids did not differ between genotypes (109.7 ± 0.93 vs. 110.6 ± 0.94 g/cm in MB and MG, respectively; *p* = 0.843), but in females, it was higher (*p* = 0.018) in MB animals (113.5 ± 0.93 g/cm) than in MG animals (109.1 ± 1.07 g/cm). From the measurements made on the cross-section of the LTL muscle, it was observed that the genotype of the kids did not affect the surface of the muscle (*p* = 0.225), but there were differences between genotypes in the shape of the muscle (*p* < 0.001). MB kids presented a lower value for the largest diameter and higher values for the minor diameter and the shape index of this muscle than MG kids. The sex of the kids only affected the width of the croup (*p* = 0.037), which was greater in females than in males.

### 3.3. Carcass Cuts

The values obtained from carcass cutting are shown in Table 4. It was observed that the genotype affected some cuts (ribs, *p* = 0.001; flanks, *p* < 0.001; tail, *p* = 0.004; kidney, *p* < 0.001) and the set of the first quality cuts of the carcass (rib + leg + shoulder, *p* = 0.014). In MB kids, the weight of the rib represented a higher percentage of the weight of the half-carcass than in MG kids, while the opposite was observed with the flanks, which were lower in MB than in MG kids. Altogether, the first quality cuts of the carcass represented a higher percentage in the crossbred MB kids than in the purebred MG kids. In the minor cuts of the carcass, it was observed that MB kids had a higher percentage of the tail and a lower percentage of the kidney than MG kids.

The sex of the kids influenced the cutting of the carcass (percentages of shoulder, *p* = 0.003; neck, *p* = 0.003; flanks, *p* = 0.003; kidney *p* = 0.043; perirenal fat, *p* = 0.001) and the set of the first quality cuts (*p* = 0.008). The carcasses of the males presented a higher percentage of shoulder and neck and a lower value of the flanks than the females. Altogether, the first quality cuts of the carcass represented a higher percentage in males than in females. In the minor cuts of the carcass, the kidney accounted for a higher percentage in males, and kidney fat was higher in females.

### 3.4. Shoulder Tissue Composition

For the tissue composition of the shoulder (Table 4), it was observed that the genotype influenced the percentage of bone (*p* < 0.001), the muscle/bone ratio (*p* < 0.001) and the meat/bone ratio (*p* < 0.001). In addition, a lower bone content, a trend of higher muscle content (*p* = 0.058) and higher ratios of muscle/bone and meat/bone were observed in MB kids compared to MG kids.

Table 4 also shows the effect of the sex of the kids on the contents of intermuscular fat, total fat and bone and muscle/bone, muscle/fat and meat/bone ratios, observing a genotype x sex interaction for intermuscular fat (*p* = 0.039). Females had a higher content of intermuscular fat than males, with the differences being more important in MB (+2.11%, *p* < 0.001) than in MG (+0.67%, *p* < 0.171). The genotype did not affect the content of intermuscular fat in males (*p* = 0.348), but MB females presented a higher value (11.42 ± 0.39%) (*p* < 0.05) than MG females (10.44 ± 0.34%) (*p* = 0.046). Total fat (subcutaneous + intermuscular) was also higher in females (*p* < 0.001), while the bone content was higher in males (*p* < 0.001), obtaining a higher muscle/fat ratio in males (*p* < 0.001) and higher muscle/bone (*p* = 0.016) and meat/bone ratios (*p* < 0.001) in the females.

### 3.5. Meat Characteristics

Table 5 shows the results of the instrumental measurements analysed for the meat quality study. It was observed that the genotype of the kids had little influence on these variables. Only some parameters of colour (a*, *p* = 0.027; C*, *p* = 0.016), ash content (*p* = 0.028) and meat firmness (*p* = 0.021) were affected. The sex of the kids did not affect the characteristics of the meat, and the genotype x sex interaction was not significant in the variables studied.

For colour, the effect of the genotype on the red index (a*) and chroma (C*) was revealed, both in the LTL muscle and in the Rectus abdominis muscle. Crossbred animals showed a higher red index and colour saturation than MG. The ash content was higher in the crossbred kids, and the firmness of the meat to the cut was slightly higher in the MG kids, indicating a greater resistance to being cut.

### 3.6. Sensory Quality of the Meat

Table 6 presents the results of the three sensory tests carried out: a panel of tasters and a controlled hedonic test of consumers on LTL muscle and a family hedonic test on leg, shoulder and rib meat. It was observed that the genotype and the sex of the kids did not affect the evaluation of the tasters or the degree of acceptance of the meat by the consumers. In the hedonic test, the other fixed factors considered did not affect the degree of acceptance of the meat either.

## 4. Discussion

### 4.1. Characteristics of the Carcass

#### 4.1.1. Genotype Effects

Pérez-Baena et al. [37] compared the visual appreciation of purebred MG kids with crossed BM of the same age but with different weights and found a greater body width and robustness for the crossed ones. When Pérez-Baena [38] compared the kids used in this experiment, with different ages but with similar live weights (9 kg), the differences were reduced, although the appearance of greater robustness for crossed kids persisted. Specifically, mestizo kids exhibited greater neck and cannon perimeters and chest width, while MG kids were slightly taller at the withers, with a larger chest girth and a slightly longer croup.

The good body morphology of Boer kids for meat production has also been highlighted by numerous authors [9,10,19,39,40], but with heavier kids than those used in the present experiment. For example, Browning et al. [10] carried out a complete diallel crossbreeding of the Boer, Kiko and Spanish breeds and found that the best morphological conformation scores, both in live animals (25–26 kg) and in carcasses (10.1–10.8 kg), were assigned to the descendants of Boer males. The descendants of Boer females also received the best rating in live animals, although this did not translate into a better carcass conformation score. However, the carcass yield was similar for all combinations of genotypes tested. Along this same line, Browning et al. [41] concluded that the genetics of the Boer breed improve the visual conformation of the kids and the carcass, which increases their market value, but it is not expected that the carcass yield (CY) values will be improved.

Teixeira et al. [42] and Sañudo et al. [43] stated that the visual evaluation of the carcass conformation in kids is highly affected by their weight, giving lower scores to kids with lower live weights. In this study, the live weight at slaughter was similar between mestizo kids and those of the MG breed, a factor that should contribute to reduced morphological differences in the carcass between genotypes.

Logically, our results showed that CY was affected by components that are not part of the carcass (feet, skin, head, viscera and digestive tract), as was likewise obtained by Panea et al. [44] and Brand et al. [45]. It was found that mestizo kids had a greater feet weight (+14.4%), which correlates with the greater cannon perimeter observed in live animals, and a greater skin weight (+10.75%), which Salehi et al. [46] correlated with the thickness of the leather obtained and Krittiya et al. [47] correlated with its quality. This greater thickness of the skin may contribute to the fact that the morphological differences between breeds are more marked in live animals than in carcasses. On the contrary, females of the MG breed had a heavier digestive tract than the rest of the animals (+10.46%) due to their greater age at slaughter, a result also obtained by Dhanda et al. [48] and Zurita-Herrera et al. [49].

CY was higher in MB females, with CCY values similar to those provided by [49] for the MG breed (50.98–52.87%). Neither Ricarte et al. [50] nor Rojas et al. [51] found differences between Criolla and Boer × Criolla kids at 6 and 10 kg live weight (46.2 vs. 46.6% and 49.9 vs. 47.1, respectively), but [51], with 15 kg kids, observed higher performance in mestizo kids (51.6 vs. 47.1, respectively).

Likewise, Stanisz et al. [16] worked with 20 kg kids of the White Improved breed and their crosses with Boers in an intensive production system, and they observed differences in CY, while [48] worked with 15 and 30 kg kids of different genotypes and their crosses with Boers in a semi-intensive production system and did not observe these differences. The results in the literature show that, in general, crossbreeding with the Boer breed does not improve CY.

On the other hand, our results showed differences in the conformation of the carcasses of MG and MB kids slaughtered at the same weight, where crossbred kids had a shorter leg length and smaller chest depth, in accordance with the shorter height and thoracic perimeter of live animals [38]. The carcass length and croup perimeter did not differ between crossbred and MG kids, but the leg compactness index was higher in crossbred kids. In kids from the Canary Goat Group, Marichal et al. [52] observed an increase in the compactness index (from 76.28 to 109.44 g/cm) when the slaughter live weight increased (from 6 to 10 kg, respectively). This conclusion seems to be confirmed since, on the one hand, [44,49] obtained lower carcass compactness indices (CCI; 84.6–107.5 g/cm) than those of this study in purebred GM kids with lower slaughter weights (6.3–7.4 kg), and on the other hand, [44] also found lower values (86.34–100.3 g/cm) for kids of five different breeds with live weights between 7.4 and 8 kg.

For the primary cuts of the carcass (leg, ribs, shoulder, flank and neck), it was observed that the values obtained in this work are very similar to those obtained in suckling kids of other breeds (Peña et al. [53], Guzmán et al. [54], Guzmán et al. [55] and Bonvillani et al. [56]), which presented mean values of 32.4–32.6%, 20.2–22.7%, 20.3–22.3 %, 9.1–10.4% and 9.3–10.5%, respectively.

The weight percentage of the rib and the total weight of this piece were higher in mestizo kids than in pure MG, although the area of the LTL muscle and the thickness of the dorsal fat did not differ statistically. However, the weight percentage of the flank (chest and abdominal wall) was higher in MG kids and was related to the greater depth of the thorax compared to mestizo kids, in agreement with the results of Santos et al. [57].

For the most valuable carcass cuts (leg + rib + shoulder), a slight superiority was observed in mestizo kids (1.16%) compared to those of the MG breed, which, for a 4.5 kg carcass, means that 52 g of meat will be sold at a higher price. This indicates that the Boer cross does not achieve a significant improvement in the proportions of valuable cuts in kids weighing 9 kg at slaughter. Likewise, Browning et al. [10] and Dhanda et al. [58] verified that the crossing of Boer males with various breeds of goats (Saanen, Angora, Kiko and Spanish) does not alter the proportions of carcass cuts in kids of 15, 25 or 30 kg.

The results obtained for tissue composition (muscle, bone, subcutaneous fat and intermuscular fat) in this work are similar to those reported by [44] in suckling MG kids: 61.9%, 24.6%, 3.1% and 9.15%, respectively. In addition, dissectible fat (DF; subcutaneous and intermuscular) did not present differences between genotypes. Dhanda et al. [58] studied the tissue composition of five genotypes, two of them mestizos from Boers with Alpina and Saanen goats, with 15 kg and 30 kg weights at slaughter. For the lighter kids, there was no clear effect of the crossbreeding with the Boer breed, while in the heavier kids, it caused a higher fat content. The tissue composition of the shoulder presented a commercial advantage for mestizo kids in this work, as it contained a lower proportion of bone (a higher proportion of meat) and higher muscle/bone and meat/bone ratios than MG kids, so it can be affirmed that the crossbred kids have a better carcass conformation.

#### 4.1.2. Sex Effects

The sex of the kids influenced the slaughter age but not the external conformation of the animals, except that the females had a greater croup width than the males. Cameron et al. [59] indicated that the visual evaluation of the carcass may not be affected by sex until the age of 11 months in crossbred Boer × Spanish kids.

On the other hand, in the process of obtaining the carcass, some differences between the sexes were observed, in that the feet, the head and the visceral organs (liver, lungs and heart) were heavier in the males. These results partially agree with [57], as no differences were found in the visceral organs, and differ from Zurita-Herrera et al. [60] and Kaić et al. [61], who found no differences between sexes in the weights of the head, skin and feet.

In the MG breed, there were no differences between sexes in the weight of NCWC nor in CY. However, in crossbred kids, NCWC was heavier in males, and CY was slightly higher in females. Males did not present differences between genotypes, but in females, MB kids reached slaughter weight at a younger age and presented a lower weight of the digestive tract and higher CY than MG kids. Most of the authors consulted [49,53,56,57] did not find significant differences in CY between males and females either. However, in Boer kids [61] and in crossbred Boer × Spanish kids [59], a slightly higher CY was found in females than in males (47.2 vs. 45.3% and 49.1 vs. 47.4%, respectively).

In the primary cuts of the carcass, some differences between sexes were observed, as the males presented a higher percentage of the neck, shoulder and kidney, as well as a lower proportion of the flank and kidney fat, than the females. The differences in the individual cuts are of little importance, although, in the carcass as a whole, the males had slightly improved (1.25%) proportions of the most valuable cuts relative to the females. The values obtained in this work are similar to those reported in suckling kids by [53,56]. However, [42,57] observed the effects of sex on some carcass cuts, although their cut methodology differed from that used in this work.

Most authors agree that males have a lower kidney fat weight, both in suckling kids [42,53,57] and in heavier kids [59,61]. However, this is not the case with other cuts. Thus, no differences were found in the weight of the neck between sexes for suckling kids [42,57], but it is higher in males than in females for heavier kids at slaughter [61]. For the weight of the shoulder of kids, Santos et al. [57] concluded that the differences vary with the breed. On the other hand, the greater weight of the flank in females, as found in this study, was also confirmed in kids with a higher live weight [59,61].

Regarding tissue composition, males had higher bone and lower fat contents than females, and, as a consequence, their carcasses had higher muscle/fat and lower muscle/bone and meat/bone ratios than females. These results agree with those obtained by [42,56], although [57] found no differences between sexes in the tissue composition of the carcasses of suckling kids.

### 4.2. Characteristics of the Meat

#### 4.2.1. Genotype Effects

The genotype of the kids had little influence on the pH of the meat, although the values of pH0 and pH24 tended to be higher in MG kids than in mestizos. The initial pH values (pH0) are similar to those observed by Zurita-Herrera et al. [62] for the MG breed (6.34–6.81), although the final pH values (pH24) are slightly higher than those obtained by [43,62,63] (5.70–5.84) for the same breed. The values shown in Table 5 are similar to those obtained by [42] in Transmontano kids and by Yalcintan et al. [64] in Saanen kids (pH0 = 6.4 and pH24 = 5.9), while Marichal et al. [52], in kids from the Canary Group, showed lower pH values (pH0 = 6.2–6.3 and pH24 = 5.59–5.73). A higher final pH is related to the stress of the animals before slaughter and affects the characteristics of the meat [13]. The small differences observed in pH between mestizo kids and MG kids in this work support the greater susceptibility to stress in MG kids noted by [43].

In this experiment, MB kids presented a higher intensity of red colour and colour saturation, differences that may be due to an effect of the genotype or feeding. Regarding the genotype, crosses with the Boer breed did not lead to conclusive results, as sometimes they improved the luminosity of the meat, while other times they did not reduce it [17,18], although they did not increase the red index [13,18] or the colour saturation C* [13]. On the other hand, the feeding of the kids in the present work was carried out using a freely available milk substitute, which had 35 mg of Fe/kg of DM. Mestizo kids had a higher growth rate and a higher daily consumption of milk than MG kids (224.2 vs. 189.1 g/d) [7]; therefore, it also represents a higher daily intake of Fe, which could have produced a slightly redder colouration of the meat. Colour is the only aspect of the meat that the consumer can evaluate in sales stalls, and they relate it to its sensory qualities [65], which is why it has great commercial importance [66].

The chemical composition of the meat was not affected by the genotype of the kids, except for the ash content, which was higher in MB. Similar values were obtained by [62,63] in suckling kids of the MG breed (moisture = 74.8–77.0%; protein = 21–22.2%; intramuscular fat = 1.02–1.97%; ashes = 1.10–1.11%), by Marichal et al. [66] and Argüello et al. [67] in kids from the Canario group, by Ripoll et al. [68] in various Spanish meat breeds and by Liotta et al. [69], but only in the ash content, for suckling Messinese kids (1.21–1.22%).

The chemical composition of muscle is relatively constant between different genotypes in its water, protein and mineral contents, but the fat content is more variable [70]. In the Boer × White Improved crossing of 20 kg animals, Stanisz et al. [16] observed that by increasing the proportion of Boer blood in mestizo kids (WI, 1/4B, 1/2B and 3/4B), the intramuscular fat (IMF) content of kids also increased (1.09, 1.14, 1.52 and 1.65, respectively). For kids of 14 to 20 kg live weight obtained from the crossing of the Boer with the Alpine breed, Brzostowski et al. [15] and Urieta et al. [71] reported a higher content of IMF compared to purebred Alpina (1.96–2.18% vs. 1.67–1.72%, respectively). Similar results were obtained by Dhanda et al. [72] for kids of 30 kg live weight from crosses between dairy and meat breeds and Boer males. Our results show that the crossbreeding of Boer males with the MG breed does not affect the composition of the meat of suckling kids with a low weight at slaughter (9 kg), but, according to the literature, in kids with a higher weight, it could lead to an increase in IMF content.

In this experiment, the muscle capacity for water retention was not affected by the genotype of the kids. Most authors have found lower values for mestizos crossed with Boers than for pure breeds, both for water loss due to meat compression [16,71,73] and for water loss due to cooking [13,73]. The capacity of the muscle to retain moisture during the cooking process plays an important role in the juiciness of the meat [74], and crossbreeding with the Boer breed either does not impair said capacity, as occurred in the present work, or rather helps to improve it, as in the cited references.

Regarding the texture of the meat, this work found that the genotype did not affect the maximum shear force (MSF), but the firmness of the meat was lower in the crossbred kids than in the pure ones, with these last values of MSF similar to those of [11] (2.55 to 5.61 kg/cm^2^) in suckling kids. Dhanda et al. [73] confirmed that the MSF (2.9 to 3.8 kg/cm^2^) did not present differences between crossbred Boer kids and other genotypes. Given the importance of meat properties related to texture, whose quality characteristics are the most appreciated by the consumer [75], the crossbred MG × Boer is interesting for its positive effects on the texture of the meat of suckling kids.

The sensory panel considered the Longissimus dorsi muscle meat to be tender and juicy, leaving little residue after chewing with a prevailing medium-intensity smell and flavour of the kid, results that coincide with those observed by [63], also in MG breed kids. Likewise, several authors who evaluated the organoleptic characteristics of kid meat [13,56,62,63,68] obtained low or medium scores for the different sensory attributes, indicating that the greater intensity of the smell and flavour of kid meat will not necessarily improve its sensory quality, particularly if consumers are not accustomed to goat meat.

In Boer kids crossbred with different breeds (Alpine, White Improved, Feral and Guanzon) with different weights (14, 20, 25 and >25 kg) and ages (50, from 72 to 95, 228 and from 180 to 300 days, respectively), a greater tenderness and juiciness of the meat was verified compared to pure breeds [13,15,16,17].

In the present work, with suckling kids (low weight and young age), no sensory differences were observed in the meat of mestizo kids or in that of the MG breed, but according to the literature information, it does not seem that crossing with the Boer breed has negative effects on the sensory quality of the meat, but rather the opposite, although these advantages could be due to the slaughter weight being higher than that in this experiment.

#### 4.2.2. Sex Effects

The sex of the kids did not affect the characteristics of the meat in this experiment, which coincides with the results of Zurita-Herrera et al. [60], also in the MG breed. In general, it is observed that the composition of the meat, the colour and the texture do not differ between males and females [41,54,57], although some breeds (Serrana and Payoya) present differences in intramuscular fat [55,57] and even in the sensory characteristics of the Serrana breed [76]. Bonvillani et al. [77] found a moderate effect of sex on the colour and the texture of the meat but not on the sensory characteristics.

## 5. Conclusions

The results of this work showed that the Murciano-Granadina × Boer mestizo kids reached slaughter weight at a younger age, were slightly shorter and had greater leg compactness and a higher Longissimus thoracis et lumborum muscle shape index, and although they did not improve the carcass yield, they had slightly improved expensive carcass cuts (particularly ribs), lower bone content and higher muscle/bone and meat/bone ratios than purebred Murciano-Granadina kids. For the meat characteristics, the genotype was of little importance, although it was observed that mestizo kids had a somewhat more intense red colour with slightly higher colour saturation and less firm meat texture. On the other hand, the males presented greater weights of the feet, head, neck, shoulder, visceral organs and kidney, while the females presented a greater width of the croup and higher proportions of the flank and kidney fat. Crossbred females presented a higher carcass yield and compactness than crossed males and purebred kids. In carcass cutting, it was observed that males had slightly improved proportions of expensive cuts, but their carcasses contained a higher proportion of bone, a lower proportion of intermuscular fat, a higher muscle/fat ratio and lower muscle/bone and meat/bone ratios than females. In no case were differences found in the sensory characteristics of the meat. It is concluded that this crossing strategy has positive effects on the commercial characteristics of the carcass and the texture of the meat, which supports the interest in crossbreeding between Murciano-Granadina females and Boer males.

## Figures and Tables

**Table 1 animals-12-02548-t001:** Least-square means (±SE) for age, slaughter weight and non-carcass weight components from kids according to genotype (purebred Murciano-Granadina (MG) or crossed with Boer (MB)) and sex.

Variables	Genotype (G)		Sex (S)		*p*-Value		
	MB (n = 50)	MG (n = 44)	Males (n = 50)	Females (n = 44)	G	S	GxS
Age (days)	44 ± 1 ^b^	63 ± 1 ^a^	48 ± 1 ^b^	59 ± 1 ^a^	<0.001	<0.001	0.051
Live weight (g)	9136 ± 57	9183 ± 57	9203 ± 53	9116 ± 56	0.553	0.262	0.470
Blood weight (g)	432 ± 19	468 ± 21	475 ± 19	426 ± 21	0.204	0.090	0.128
Skin weight (g)	1143 ± 9 ^a^	1032 ± 10 ^b^	1098 ± 9	1077 ± 10	<0.001	0.134	0.426
Feet weight (g)	397 ± 3 ^a^	347 ± 3 ^b^	393 ± 3 ^a^	350 ± 3 ^b^	<0.001	<0.001	0.906
Head weight (g)	470 ± 3	474 ± 3	479 ± 3 ^a^	466 ± 3 ^b^	0.443	0.006	0.096
Viscera weight (g)	537 ± 5	538 ± 6	546 ± 5 ^a^	526 ± 6 ^b^	0.826	0.019	0.795
Gastrointestinal tract (g)	1220 ± 28	1389 ± 30	1246 ± 28	1362 ± 30	<0.001	0.006	0.003

^a,b^ Different superscripts in the same row and factor indicate significant differences at *p* < 0.05.

**Table 2 animals-12-02548-t002:** Least-square means (±SE) for total non-carcass weight components, carcass weight and dressing percentage from kids according to genotype (purebred Murciano-Granadina (MG) or crossed with Boer (MB)) and sex.

Variables	MB		MG		*p*-Value		
	Males (n = 25)	Females (n = 25)	Males (n = 25)	Females (n = 19)	G ^1^	Sex	GxSex
NCWC (g)	4254 ± 34 ^a^	4148 ± 34 ^b^	4219 ± 34 ^a^	4270 ± 39 ^a^	0.224	0.432	0.029
HCW (g)	4761 ± 32 ^a^	4843 ± 32 ^a^	4732 ± 32 ^ab^	4673 ± 37 ^b^	0.004	0.732	0.037
CCW (g)	4616 ± 31 ^b^	4725 ± 31 ^a^	4624 ± 31 ^b^	4537 ± 35 ^b^	0.006	0.744	0.003
HCY (%)	52.10 ± 0.35 ^ab^	53.00 ± 0.35 ^a^	51.77 ± 0.35 ^b^	51.10 ± 0.40 ^b^	0.003	0.757	0.035
CCY (%)	50.53 ± 0.34 ^b^	51.71 ± 0.34 ^a^	50.60 ± 0.34 ^b^	49.62 ± 0.39 ^b^	0.005	0.782	0.003

^1^ Genotype; NCWC: non-carcass weight components; HCW: hot carcass weight; CCW: cold carcass weight; HCY: hot carcass yield; CCY: cold carcass yield. ^a,b^ Different superscripts in the same row indicate significant differences at *p* < 0.05.

**Table 3 animals-12-02548-t003:** Least-square means (±SE) for carcass measurements from kids according to genotype (purebred Murciano-Granadina (MG) or crossed with Boer (MB)) and sex.

Variables	Genotype (G)		Sex (S)		*p*-Value		
	MB (n = 50)	MG (n = 44)	Males (n = 50)	Females (n = 44)	G	S	GxS
Measurements on the carcass:							
Croup width (cm)	13.69 ± 0.08 ^b^	14.06 ± 0.08 ^a^	13.75 ± 0.08 ^b^	13.99 ± 0.08 ^a^	0.002	0.037	0.181
Croup perimeter (cm)	35.67 ± 0.12	35.94 ± 0.13	35.66 ± 0.12	35.95 ± 0.13	0.128	0.112	0.577
Leg length (cm)	21.53 ± 0.12 ^b^	22.36 ± 0.13 ^a^	21.85 ± 0.12	22.02 ± 0.13	<0.001	0.322	0.812
Internal length of the carcass (cm)	41.87 ± 0.18	41.72 ± 0.19	41.97 ± 0.18	41.62 ± 0.19	0.549	0.188	0.567
Chest depth (cm)	17.13 ± 0.10 ^b^	17.71 ± 0.10 ^a^	17.29 ± 0.10	17.55 ± 0.10	<0.001	0.082	0.779
Leg compactness index (g/cm)	36.28 ± 0.27 ^a^	34.71 ± 0.29 ^b^	35.85 ± 0.28	35.20 ± 0.30	<0.001	0.193	0.188
Carcass compactness index (g/m)	111.61 ± 0.65	109.84 ± 0.71	110.15 ± 0.66	111.29 ± 0.71	0.071	0.245	0.006
Measurements on the Longissimus thoracis et lumborum muscle:							
Largest diameter (cm)	3.85 ± 0.06 ^b^	4.23 ± 0.07 ^a^	4.03 ± 0.06	4.05 ± 0.07	<0.001	0.827	0.360
Minor diameter (cm)	1.93 ± 0.05 ^a^	1.71 ± 0.05 ^b^	1.85 ± 0.05	1.80 ± 0.05	0.005	0.566	0.562
Fat thickness (cm)	0.41 ± 0.03	0.34 ± 0.03	0.41 ± 0.03	0.35 ± 0.03	0.110	0.150	0.507
Muscle area (cm^2^)	5.64 ± 0.95	5.40 ± 0.85	5.59 ± 0.92	5.46 ± 0.87	0.225	0.507	0.387
Muscle shape index	50.88 ± 1.65 ^a^	40.66 ± 1.71 ^b^	46.57 ± 1.58	44.98 ± 1.77	<0.001	0.507	0.387

^a,b^ Different superscripts in the same row and factor indicate significant differences at *p* < 0.05.

**Table 4 animals-12-02548-t004:** Least-square means (±SE) for the carcass cuts as a percentage of the half carcass and for back-tissue composition in kids according to genotype (purebred Murciano-Granadina (MG) or crossed with Boer (MB)) and sex.

Variables	Genotype (G)		Sex (S)		*p*-Value		
	MB	MG	Males	Females	G	S	GxS
Carcass cuts (%):	(n = 50)	(n = 44)	(n = 50)	(n = 44)			
Kidney	1.12 ± 0.02 ^b^	1.28 ± 0.02 ^a^	1.23 ± 0.02 ^a^	1.17 ± 0.02 ^b^	<0.001	0.043	0.378
Perirenal fat	3.00 ± 0.13	2.81 ± 0.14	2.58 ± 0.13 ^b^	3.23 ± 0.14 ^a^	0.306	0.001	0.590
Tail	0.73 ± 0.02 ^a^	0.63 ± 0.02 ^b^	0.66 ± 0.02	0.70 ± 0.02	0.004	0.166	0.489
Shoulder	21.28 ± 0.15	21.03 ± 0.16	21.50 ± 0.15 ^a^	20.81 ± 0.16 ^b^	0.274	0.003	0.256
Leg	33.48 ± 0.21	33.66 ± 0.22	33.78 ± 0.21	33.36 ± 0.22	0.546	0.167	0.305
Rib	21.18 ± 0.22 ^a^	20.08 ± 0.24 ^b^	20.70 ± 0.23	20.56 ± 0.22	0.001	0.655	0.914
Flank	10.06 ± 0.16 ^b^	10.97 ± 0.17 ^a^	10.16 ± 0.16 ^b^	10.87 ± 0.17 ^a^	<0.001	0.003	0.522
Neck	8.45 ± 0.12	8.51 ± 0.13	8.77 ± 0.12 ^a^	8.20 ± 0.13 ^b^	0.745	0.003	0.793
First quality cuts ^1^	75.93 ± 0.31 ^a^	74.77 ± 0.34 ^b^	75.98 ± 0.31 ^a^	74.73 ± 0.34 ^b^	0.014	0.008	0.254
Back-tissue composition (%) (n = 30):							
Muscle	61.78 ± 0.27	61.04 ± 0.27	61.70 ± 0.27	61.12 ± 0.27	0.058	0.132	0.185
Subcutaneous fat	2.94 ± 0.14	3.07 ± 0.14	2.88 ± 0.14	3.13 ± 0.14	0.488	0.196	0.359
Intermuscular fat	10.37 ± 0.24	10.11 ± 0.24	9.54 ± 0.24 ^b^	10.93 ± 0.24 ^a^	0.444	<0.001	0.039
Total fat	13.30 ± 0.24	13.17 ± 0.24	12.42 ± 0.24 ^b^	14.06 ± 0.24 ^a^	0.707	<0.001	0.120
Bone	21.18 ± 0.17 ^b^	22.15 ± 0.17 ^a^	22.16 ± 0.17 ^a^	21.17 ± 0.17 ^b^	<0.001	<0.001	0.737
Muscle/bone	2.93 ± 0.03 ^a^	2.76 ± 0.03 ^b^	2.80 ± 0.03 ^b^	2.89 ± 0.03 ^a^	<0.001	0.016	0.940
Muscle/fat	4.72 ± 0.10	4.70 ± 0.10	5.01 ± 0.10 ^a^	4.41 ± 0.10 ^b^	0.872	<0.001	0.128
Meat/bone	3.56 ± 0.04 ^a^	3.36 ± 0.4 ^b^	3.36 ± 0.04 ^b^	3.56 ± 0.04 ^a^	<0.001	<0.001	0.601

^1^ First quality cuts = shoulder + leg + ribs. ^a,b^ Different superscripts in the same row and factor indicate significant differences at *p* < 0.05.

**Table 5 animals-12-02548-t005:** Least-square means (±SE) for meat characteristics from kids according to genotype (purebred Murciano-Granadina (MG) or crossed with Boer (MB)) and sex.

Variables	Genotype (G)		Sex (S)		*p*-Value		
	MB	MG	Males	Females	G	S	GxS
	(n = 50)	(n = 44)	(n = 50)	(n = 44)			
pH0	6.47 ± 0.03	6.56 ± 0.04	6.52 ± 0.03	6.50 ± 0.03	0.073	0.711	0.258
pH45	6.18 ± 0.03	6.22 ± 0.04	6.20 ± 0.04	6.20 ± 0.04	0.491	0.863	0.972
pH24	5.85 ± 0.03	5.92 ± 0.03	5.88 ± 0.03	5.89 ± 0.03	0.083	0.923	0.557
Colour: Longissimus thoracis muscle:							
Brightness (L*)	47.75 ± 0.42	48.05 ± 0.45	48.23 ± 0.42	47.57 ± 0.45	0.628	0.287	0.914
Red index (a*)	8.57 ± 0.25 ^a^	7.74 ± 0.27 ^b^	8.15 ± 0.25	8.16 ± 0.27	0.027	0.977	0.321
Yellow index (b*)	2.77 ± 0.10	2.49 ± 0.13	2.67 ± 0.12	2.60 ± 0.13	0.115	0.755	0.290
Chroma (C*)	9.06 ± 0.24 ^a^	8.20 ± 0.26 ^b^	8.63 ± 0.24	8.63 ± 0.26	0.016	0.998	0.418
Tone (H^o^)	18.61 ± 0.99	18.16 ± 1.07	18.73 ± 0.99	18.05 ± 1.07	0.759	0.641	0.146
Colour: Rectus abdominis muscle:							
Brightness (L*)	52.83 ± 0.36	53.70 ± 0.38	53.24 ± 0.36	53.29 ± 0.38	0.096	0.923	0.537
Red index (a*)	8.34 ± 0.25 ^a^	7.38 ± 0.26 ^b^	7.63 ± 0.26	8.08 ± 0.25	0.009	0.213	0.240
Yellow index (b*)	1.60 ± 0.22	1.50 ± 0.24	1.55 ± 0.22	1.55 ± 0.24	0.762	0.999	0.600
Chroma (C*)	8.64 ± 0.24 ^a^	7.72 ± 0.25 ^b^	8.40 ± 0.24	7.79 ± 0.25	0.009	0.217	0.277
Tone (H^o^)	11.72 ± 1.75	12.00 ± 1.87	11.91 ± 1.76	11.81 ± 1.86	0.910	0.970	0.737
Chemical composition of meat (%) (n = 30):							
Humidity	76.90 ± 0.21	77.06 ± 0.24	77.25 ± 0.23	76.71 ± 0.24	0.089	0.522	0.304
Protein	21.01 ± 0.38	20.79 ± 0.30	21.18 ± 0.32	20.63 ± 0.37	0.645	0.272	0.575
Fat	1.41 ± 0.05	1.31 ± 0.07	1.27 ± 0.05	1.45 ± 0.07	0.228	0.055	0.182
Ashes	1.26 ± 0.03	1.13 ± 0.04	1.23 ± 0.03	1.15 ± 0.03	0.028	0.120	0.986
Water retention capacity (WRC) and texture (n = 30):							
WRC (%)	21.35 ± 0.92	22.03 ± 1.10	21.89 ± 0.92	21.58 ± 1.05	0.637	0.870	0.118
Lost cooked (%)	18.92 ± 0.92	20.50 ± 0.94	19.15 ± 0.99	20.27 ± 0.87	0.245	0.408	0.127
Strength (kg/cm^2^)	4.59 ± 0.20	4.97 ± 0.20	4.78 ± 0.21	4.79 ± 0.19	0.212	0.977	0.785
Firmness (kg/cm^2^)	1.95 ± 0.08 ^b^	2.23 ± 0.08 ^a^	2.09 ± 0.08	2.09 ± 0.08	0.021	0.956	0.537
Total work (kg/cm^2^)	8.50 ± 0.2	8.94 ± 0.3	8.86 ± 0.45	8.58 ± 0.40	0.487	0.656	0.775

^a,b^ Different superscripts in the same row and factor indicate significant differences at *p* < 0.05.

**Table 6 animals-12-02548-t006:** Least-square means (±SE) for the sensory quality of kid meat according to genotype (purebred Murciano-Granadina (MG) or crossed with Boer (MB)) and sex.

Variable	Genotype (G)		Sex (S)		*p*-Value		
	MB	MG	Males	Females	G	S	GxS
Evaluation by a panel of tasters (Longissimus dorsi thoracis muscle) ^1^:							
Kid odour intensity	50.19 ± 1.34	48.50 ± 1.37	49.94 ± 1.40	49.75 ± 1.31	0.728	0.920	0.499
Grease odour intensity	30.08 ± 0.81	30.04 ± 0.78	30.36 ± 0.74	29.76 ± 0.84	0.970	0.595	0.068
Kid flavour	43.45 ± 1.24	40.59 ± 1.26	43.13 ± 1.29	40.91 ± 1.21	0.113	0.215	0.129
Liver flavour	21.03 ± 1.21	18.44 ± 1.24	20.25 ± 1.26	19.23 ± 1.18	0.143	0.560	0.780
Metallic flavour	22.89 ± 0.97	21.06 ± 0.98	22.43 ± 0.98	21.52 ± 0.97	0.192	0.511	0.675
Sour flavour	15.62 ± 1.06	13.51 ± 1.07	14.82 ± 1.09	14.32 ± 1.04	0.167	0.743	0.804
persistence	38.11 ± 1.04	36.92 ± 1.06	38.28 ± 1.08	36.76 ± 1.02	0.426	0.312	0.205
Tenderness	55.00 ± 2.40	54.19 ± 2.47	56.21 ± 2.57	52.98 ± 2.30	0.813	0.354	0.727
Juiciness	44.66 ± 1.10	43.43 ± 1.11	44.71 ± 1.14	43.38 ± 1.07	0.438	0.400	0.955
Fibrosity	35.09 ± 1.84	36.49 ± 1.90	35.23 ± 1.96	36.35 ± 1.77	0.597	0.674	0.511
Residue amount	32.75 ± 1.68	34.15 ± 1.73	33.27 ± 1.79	33.64 ± 1.62	0.563	0.875	0.470
General acceptance	49.80 ± 1.53	48.85 ± 1.57	50.33 ± 1.62	48.31 ± 1.49	0.668	0.365	0.904
Controlled consumer hedonic testing (Longissimus thoracis et lumbarum muscle) ^2^:							
Acceptance degree	8.51 ± 0.33	7.92 ± 0.33	8.21 ± 0.33	8.22 ± 0.33	0.238	0.976	0.322
Familiar hedonic testing (leg, shoulder and rib) ^2^:							
Acceptance degree	9.94 ± 0.49	9.79 ± 0.49	9.75 ± 0.49	9.98 ± 0.49	0.550	0.369	0.630

^1^ Score on a 100-point scale; ^2^ score on a 15-point scale.

## Data Availability

Not applicable.

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
