# Peer review of "Terminal Crossbreeding of Murciano-Granadina Goats with Boer Bucks: Characteristics of the Carcass and the Meat"

_animals, 2022, doi:10.3390/ani12192548_

Round 1

Reviewer 1 Report

The researchers have done well at describing the research.  There were limited differences so the author might consider combining some of the information.  There seems to be problems with consistency of data presentation and some of the references.  I have specific comments in the attached file.

Author Response

Manuscript title: Terminal Crossbreeding of Murciano-Granadina Goats to Boer Bucks: Characteristics of the Carcass and the Meat

[Animals] Manuscript ID: animals-1863270

Reviewer 1:

All comments have been considered. Those that do not appear below have been modified as noted by the reviewer. The modifications made have been highlighted in yellow for better location.

Former line 77: This live animal management was used by the authors of reference 7 of this manuscript, therefore in this work only the kids obtained in said reference 7 were used.

Former lines 143-144:  You are right, in sheep and goats a blade speed of 200 mm/min is used, although in the review prior to the start-up of this experiment we also found that in smaller animals, higher blade speed values are usually applied (300mm/min or 5mm/s), so we consider that 5mm/s would be suitable for kids of low age and live weight. Some references for this parameter can be seen in:  

  • Ramı́rez, A.; Olivera, M.A.; Pla, M.; Guerrero, L.; Ariño, B.; Blasco, A.; Pascual, M.; Gila, M. Effect of selection for growth rate on biochemical, quality and texture characteristics of meat from rabbits. Meat Science. 2004, 67: 4, 617-624. https://doi.org/10.1016/j.meatsci.2003.12.012.
  • Ariño, A.; Hernández, P.; Blasco, A. Comparison of texture and biochemical characteristics of three rabbit lines selected for litter size or growth rate. Meat Science. 2006, 73: 4, 687-692. https://doi.org/10.1016/j.meatsci.2006.03.014.

Table 3: The structure of tables 1 and 3 is identical; in them, the genotype is compared on the one hand and the sex on the other. There is only one significant interaction in each table, which is explained in the text. However, Table 2 presents a different structure, which allows us to observe the significant Genotype x Sex interactions in all the variables analysed. The design difference in the tables is due to the fact that we considered it more important to highlight the effect of genotype and sex, when the interaction is mostly non-significant in the different variables considered.

Reviewer 2 Report

The manuscript deals with the crossbreeding between Murciano-Granadina and Boer to improve goats' carcass and meat quality. This manuscript is interesting and well writing. However, the analysis model should be reconsidered and reanalyzed data. There are some comments and suggestions as follows:

Line 31: The authors needed to provide full name of “MB” before using the abbreviation in the abstract.

Lines 31 – 36: Please provide quantitative values (i.e.,  Lease square means and SE, and p-value) to support the sentences.

Lines 21 – 90: Did the author request permission to use the animals for scientific experiments? Please provide the information.

Line 186: Should “breed” mentioned in the model be “genotype”? Due to the relevant content, such as the models in lines 189, 192, and 194, and reported results, especially in tables, the authors use the term “genotype”.  However, I think using “breed” would be more appropriate than “genotype” to use across the manuscript.

Line 197: What method was used for comparing the multiple means? Please provide information.

Lines 185 – 186: Different slaughtered age was carried out between MB and MG breeds, so age at slaughter should be included in the model to analyze for carcass traits and weight traits (liver, blood, skin, feet, head viscera, and gastro-intestinal tract) as covariate fixed effect. The authors were suggested to reanalyze the data.

Lines 202 – 205: Least square means and SE needed to be provided to support the sentences.

Tables 1 – 6: Least square means ± SE (standard error) should be reported instead of using  Least square means ± SEM (standard error of the mean).

Table 4 – 5: Please recheck and change the “,” to “.”.

Table 4: Please recheck the number of tissue compositions (n = 30). The authors mentioned that only 15 animals of each type were used for this study.

Lines 292 – 298 and Table 6: This was the balance data analysis; thus, mean and SD should be appropriated to report in this section instead of using Lease square means and SE. 

Author Response

Manuscript title: Terminal Crossbreeding of Murciano-Granadina Goats to Boer Bucks: Characteristics of the Carcass and the Meat

[Animals] Manuscript ID: animals-1863270

Reviewer 2:

All comments have been considered. Those that do not appear below have been modified as noted by the reviewer. The modifications are highlighted in green for better location.

Former lines 21-90: The animals used in this work were fattened by Pérez-Baena et al. [7] and slaughtered later.  A letter from the Ethics and Animal Welfare Committee from our university has been submitted to the editor stating that the handling of animals carried out in this work does not require any permission from said committee.

Former line 186: We prefer the use of "genotype" because we think this concept is more appropriate than "breed" for MG x Boer animals.

Former line 185-186: At least, in the Spanish market, the weight of the kids is considered more important than their age. Given the difference in growth of the kids between genotypes (MB > MG) in the intensive system, slaughter at the same age would lead to very different weights in MB and MG, and the MB could even fall outside the commercial weight range (7 -9 kg) and suffer depreciation of the carcass. We consider that introducing the age of the animals as a covariate in the statistical model would give results that would not be practical for the sector, so we believe that it is better to leave the model as it is.

Former line 202-205:  Journal editors routinely instruct authors not to repeat data in the text that is available in tables, and this is what we have done in this manuscript. The appearance of data in lines 206-209 is due to the existence of an interaction whose values do not appear in the tables.

Tables 1-6: You are right. We have changed the SEM to SE in all the tables

Table 4: The number of animals in Table 4 (tissue composition) is correct, since each genotype groups 15 males and 15 females. On the other hand, each sex grouped 15 animals of each genotype (15 MG and 15 MB).

Former lines 292-298 and Table 6:  Please see Tables 1-6 answer. It must be taken into account that although the number of animals per factor is the same, the high number of variables and samples analysed means that, in some cases, some of these samples may have been lost or damaged, giving a different SE.

Reviewer 3 Report

Reviewer(s)’ General Comments to Authors:

The manuscript is overall well written, informative and the results reported look suggestive, but the quality of the paper needs improvement. Minor suggestions are proposed prior to publication. I believe that the topic is interesting, and the results are of value to goat farmers and the wider research community. A few sentences did not make sense and the paper requires a bit of re-working. References could be supplemented.

Line 53: …with an  average milk…

Line 60: … in goats consists of crossing…

Line 64: … showed that the crossbreeding…

Line 114: … and the results were expressed…

Line 184: The statistical part could be improved, and explained a little better, information on some analyses and statistical methods is missing…

Line 249: … the flanks, that which were lower…

Line 269: Table 4 also shows an the effect…

Line 291: … cating a greater resistance…

Line 306: … with different weights

Line 319: … translate into a better…

Line 330: … that CY was affected…

Line 346: … worked with kids..

Line 404: … NCWC was heavier…

Line 421: … were found for the weight… Could it be possible to rephrase the sentence?

Line 443: … susceptibility to the stress of…

Line 488: … crossbreed it is…

Line 494: … obtain scores of low…

Some results should be discussed in more detail with possible explanations. Moreover, the conclusion could contain more comments to be more specific and clear for the reader of what your study showed eventually, with more detailed information but summarized.  

Author Response

Manuscript title: Terminal Crossbreeding of Murciano-Granadina Goats to Boer Bucks: Characteristics of the Carcass and the Meat

[Animals] Manuscript ID: animals-1863270

Reviewer  3:

All comments have been considered and taken into account.

Round 2

Reviewer 2 Report

Dear Editor,

The revision and the information provided by the authors are acceptable. So, this version of the manuscript would be recommended for publication.

Best regards.

Author Response

OK.